# Predictive Models of within- and between-Species SARS-CoV-2 Transmissibility

**DOI:** 10.3390/v14071565

**Published:** 2022-07-19

**Authors:** Ricardo Soares, Cristina P. Vieira, Jorge Vieira

**Affiliations:** 1Faculdade de Ciências, Universidade do Porto (FCUP), Rua do Campo Alegre, s/n, 4169-007 Porto, Portugal; ricardo.soares@i3s.up.pt; 2Instituto de Biologia Molecular e Celular (IBMC), Rua Alfredo Allen, 208, 4200-135 Porto, Portugal; cgvieira@ibmc.up.pt; 3Instituto de Investigação e Inovação em Saúde (I3S), Universidade do Porto, Rua Alfredo Allen, 208, 4200-135 Porto, Portugal

**Keywords:** SARS-CoV-2, Coronaviridae, spike glycoprotein (S), angiotensin converting enzyme 2 (ACE2), prediction model, interfacing residues

## Abstract

Viruses from the *Coronaviridae* family have been reported to infect a large range of hosts, including humans. The latest human-infecting coronavirus, SARS-CoV-2, turned into a pandemic and subtypes with different transmissibility have appeared since then. The SARS-CoV-2 Spike (S) protein interacts with the angiotensin-converting enzyme 2 (ACE2) host receptor, and thus, in silico models, based on the structural features of the SARS-CoV-2 S protein–ACE2 receptor complex, as well as ACE2 amino acid patterns, may be used to predict the within- and between-species transmissibility of SARS-CoV-2 subtypes. Here, it is shown that at the beginning of the pandemic, the SARS-CoV-2 S protein was, as expected for a virus that just jumped the species barrier, ill-adapted to the human ACE2 receptor, and that the replacement of one SARS-CoV-2 variant by another is partially due to a better fitting of the S protein–human ACE2 complex. Moreover, it is shown that mutations that are predicted to lead to a better fit have increased in the population due to positive selection. It is also shown that the number of ACE2-interfacing residues is positively correlated with the transmissibility rate of SARS-CoV-2 variants. Finally, it is shown that the number of species that are susceptible to infection by SARS-CoV-2, and that could be a reservoir for this virus, is likely higher than previously thought.

## 1. Introduction

Coronaviruses (CoVs) are enveloped viruses with a positive-sense single-stranded RNA genome and a nucleocapsid of helical symmetry, that in humans can cause respiratory tract infections that range from mild to lethal. They were first described in patients diagnosed with the common cold [1], and can be classified into four genera, namely α-, β-, δ-, and γ-coronavirus [2]. Nevertheless, only seven species from the α- and β -coronavirus genera are known to infect humans, namely HCoV-229E and HCoV-NL63, which belong to the former genus and HCoV-HKU1, HCoV-OC43, severe acute respiratory syndrome coronavirus (SARS-CoV), Middle East respiratory syndrome coronavirus (MERS-CoV), and severe acute respiratory syndrome coronavirus 2 (SARS-CoV-2) which belong to the latter [3]. Three (SARS-CoV, MERS-CoV, and SARS-CoV-2) out of the seven known coronaviruses that sicken people cause a severe disease. SARS-CoV and MERS-CoV had a relatively low number of infections (few thousands) and showed high mortality rates, 10% and 34.4%, respectively [4,5]. In contrast, by the end of 2020, SARS-CoV-2 had 80 million infection cases and a mortality rate of 2.2% [6].

The spike glycoprotein (S), one of the four structural proteins encoded by CoVs, mediates the cell tropism and host range of the CoV virion [7,8]. This protein forms homotrimers that protrude through the viral surface, with each protomer consisting of a single polypeptide chain, typically with a minimum of 1100 residues to a maximum of 1600 [8].

The S protein must be processed by host proteases at the S2’ fusion site to generate two functional subunits, designated S1 and S2. This event leads to membrane fusion and viral entry into the cell [8,9,10,11]. The S1 subunit constitutes the trimers’ apex and includes the receptor-binding domain (RBD), responsible for binding to host cell receptors, but it also stabilizes the S2 fusion machinery while in a prefusion state. The S2 subunit is anchored in the viral membrane and mediates membrane fusion, enabling CoVs entry into the host [7,8].

In contrast to other CoVs, MERS-CoV, SARS-CoV, and SARS-CoV-2 show different states for the B subdomain of the S1 subunit. The closed state is stable and receptor inaccessible, while the opened state is less stable and receptor accessible. For these three viruses, the latter conformation is essential for engaging the cell receptors [12,13]. Usually, the S protein homotrimers of these viruses are in a partially-opened conformation [14].

There are four main protein receptors that CoVs can use to bind to host cells [2], namely amino peptidase N (APN), murine carcinoembryonic antigen-related cell adhesion molecule 1 (CEACAM1), dipeptidyl peptidase 4 (DPP4), and angiotensin-converting enzyme 2 (ACE2). Phylogenetically-related species tend to use the same receptor, but there are exceptions. For instance, both HCoV-NL63 (which belongs to the α-coronavirus genus) and SARS-CoV-2 (which belongs to the β- coronavirus genus) use the ACE2 receptor, although all other species of the α-coronavirus genus, including one that is able to infect humans (HCoV-229E), use the APN receptor. Knowing which kind of receptor is used by a given CoV is important for predicting transmissibility rates using in silico approaches. Recently, Carvalho and Alves [15] reported that the S protein hydrophobic motif YGFY is an indicator of ACE2 usage. Nevertheless, this sequence is not present in SARS-CoV-2, where the YGFQ sequence is present. Likewise, the Bat SARS-like CoV Rs7327 which uses ACE2, also presents the YGFF sequence which is identical to the one presented by the Bat CoV BM48–31, a species that is documented as being unable to use the ACE2 receptor [16]. Here, we describe an amino acid pattern (CYX(6)GX(3)T[^V] at positions 488–501 of SARS-CoV-2 S protein) that identifies all β-coronavirus species known to use ACE2 with the exception of HCoV-NL63, which does not belong to the B lineage, and that is not present in CoV species that do not use ACE2.

SARS-CoV-2 enters the host cells by likely binding two trimeric S protein structures with an ACE2 homodimer [17]. This is the single entry point of SARS-CoV-2 since HeLa cells not expressing ACE2 cannot be infected by SARS-CoV-2 [18]. This result shows that ACE2 is required for cell entry but does not exclude other factors, such as carbohydrate binding, as being important as well (see for instance [19]). ACE2 is an integral membrane protein that is expressed in the heart, lungs, kidneys, and intestines, and is a key player in the renin–angiotensin system [17,20]. ACE2 acts as an ACE antagonist by inactivating angiotensin II and thereby reducing blood pressure. Both genes can be found in vertebrates, primitive tunicates, and chordates [21,22]. ACE and ACE2 are evolutionary related since they are the result of an old gene duplication event [23]. The Human ACE2 is 805 amino acids long and it possesses an extracellular domain and a cytoplasmic tail. Three extracellular domain regions (30–41, 82–84, and 353–357) have been shown to interact with the SARS-CoV-2 spike glycoprotein [18]. This is the region of the N-terminal zinc metallopeptidase domain (PD; residues 19–611), which cleaves Ang I, yielding Ang-(1–9) [24].

The SARS-CoV-2 S protein receptor-binding domain (RBD) is essential for the interaction with ACE2. Within the RBD, the receptor-binding motif (RBM) includes 16 out of the 17 residues that interact with the 20 ACE2 critical residues [18]. Residue 417, located outside the RBM, also interacts with ACE2. The SARS-CoV-2 contacting residues are: K417, G446, Y449, Y453, L455, F456, A475, F486, N487, Y489, Q493, G496, Q498, T500, N501, G502, and Y505. For ACE2, the residues are: Q24, T27, F28, D30, K31, H34, E35, E37, D38, Y41, Q42, L79, M82, Y83, N330, K353, G354, D355, R357, and R393 [18].

The Centers for Disease Control and Prevention (CDC) established a three class classification of SARS-CoV-2 variants based on their threat level: variants of interest (VOI), variants of concern (VOC), and variants of high consequence (VHC). In order to classify a variant as a VOC rather than as a VOI, evidence for increased transmissibility or disease severity when compared to the wild type must be available. If those increases are severe, then it becomes a VHC. By the end of January 2021, several SARS-CoV-2 strains had been detected in a large number of countries, of which we highlight VOCs B.1.1.7 or 20I/501Y.V1, B.1.351 or 20H/501Y.V2, P.1 or 20J/501Y.V3 and VOI B.1.429 or 20C/S:452R, which show amino acid changes in their S protein and a confirmed increase in transmissibility rate compared to SARS-CoV-2 wild type [25,26,27,28].

Having a computational method that can be used to predict the relative transmissibility of SARS-CoV-2 strains, as here reported, is desirable, since it allows policy makers to take action before a variant with increased transmissibility spreads throughout the population. As expected, we also show that variants classified as VOIs and VOCs have increased in frequency in the population due to positive selection.

SARS-CoV, MERS-CoV, HCoV-229E, and HCoV-NL63 are thought to have a bat origin, while HCoV-OC43 and HCoV-HKU1 may have reservoirs in rodents or cows [2,29,30]. In the case of MERS-CoV and SARS-CoV, dromedaries and civets respectively, have been identified as putative intermediate hosts [31,32]. As for SARS-CoV-2, it is hypothesized that it had a zoonotic origin in bats or pangolins [33]. Nevertheless, the intermediate hosts are unknown [34]. Given the evidence that CoVs can cross species boundaries, determining which species can act as SARS-CoV-2 reservoirs is of interest.

Damas et al. [35] used an ACE2 sequence dataset from 410 vertebrate species, including 252 mammals, to study the potential of ACE2 to be used as a receptor by SARS-CoV-2. These authors looked at the conservation properties of 25 amino acids important for the binding of ACE2 to the SARS-CoV-2 S protein to develop a five-category binding score. They also employed a protein structural analysis to qualitatively assess whether amino acid changes at these residues would be likely to disrupt ACE2/SARS-CoV-2 S protein binding, and found that the number of predicted unfavorable changes significantly correlated with the binding score. In their scoring scheme, only mammals fell into the medium to very high categories and only catarrhine primates into the very high category. Although useful, this model is certainly not perfect since bats and civets were not identified as potential SARS-CoV-2 reservoirs. Here, we also present a 11 residues ACE2 motif, based on the amino acid characteristics observed in susceptible species, that can be used to predict which species are susceptible to SARS-CoV-2, and that in opposition to the scoring scheme of [35] is able to predict bats and civets as susceptible species.

## 2. Materials and Methods

### 2.1. Sequence Retrieval

The acquisition of sequences started by querying for both “Animals” and “Coronaviridae” in the NCBI Assembly RefSeq and Assembly GenBank databases, respectively. CDS annotations were downloaded in FASTA (.fna) format. The SARS-CoV-2 wild type S protein sequence (YP_009724390.1) was extracted from the reference genome deposited in NCBI (NC_045512.2). Using SEDA [36], a tblastn (the chosen expect value was 0.05) was performed using each animal CDS and *Coronaviridae* CDS as the subject and hACE2 (NP_001358344.1) and SARS-CoV-2 S protein (YP_009724390.1) as the query, respectively. A Docker image is available for SEDA at the pegi3S Bioinformatics Docker Images Project (https://pegi3s.github.io/dockerfiles/; accessed on 1 November 2021; [37]).

To ensure that all animal sequences belonged to the ACE2 protein, the retrieved sequences were aligned (ClustalW codons), and a Neighbor-joining phylogenetic tree (bootstrap value of 500) was obtained using MEGAX [38].

The ViPR was also queried for more S protein data. Searching Coronaviridae family and querying for “spike glycoprotein”, all sequences associated with complete genome annotation were downloaded. The retrieved sequences were processed as described in the general “Preparing datasets for large scale phylogenetic analyses protocol” (https://www.sing-group.org/seda/manual/index.html; accessed on 1 November 2021), using the dataset builder software SEDA [36]. From this dataset, a blastp was performed to determine the 10 S proteins most similar to SARS-CoV-2.

Simultaneously, lists for positive and negative controls of SARS-CoV-2 infection were made. The former consisted of 12 species (*Homo sapiens*, *Felis catus*, *Macaca fascicularis*, *Macaca mulatta*, *Mustela putorius furo*, *Peromyscus maniculatus*, *Rousettus aegyptiacus*, *Mesocricetus auratus*, *Callithix jacchus*, *Panthera leo*, *Sus scrofa*, *Neovison vison,* and *Mus musculus*), and the latter of five species (*Anas platyrhynchos domesticus*, *Gallus gallus*, *Anser cygnoides*, *Coturnix japonica*, and *Meleagris gallopavo f. domestica*) [39,40,41]. ACE2 sequences from species not present in the dataset retrieved from NCBI were gathered by searching for ACE2 orthologs in Ensembl (https://www.ensembl.org/index.html; accessed on 1 November 2021).

For SARS-CoV-2 variants, their set of mutations were gathered, preferentially from pangolin (https://cov-lineages.org/resources/pangolin.html; [42]), complemented with information from the CDC (Centers for Disease Control and Prevention [43]). Sets of mutations were individually edited in the FASTA file containing the SARS-CoV-2 reference S protein (YP_009724390.1; see Table 1).

### 2.2. Coronaviridae’s Use of ACE2

Sequences from species with closely-related S proteins to SARS-CoV-2, as determined by blastp, were aligned to enable the search of sequence patterns related to the use of ACE2. Furthermore, S protein sequences documented to use or not to use ACE2 as a receptor were added. From these sequences, several protein 3D models were obtained using the Iterative Threading ASSEmbly Refinement (I-TASSER) web server (http://zhanglab/I-TASSER/; accessed on 1 November 2021; [45]) and their structure trimmed to only contain the receptor-binding motif [18]. These RBMs were aligned with SARS-CoV-2’s RBM using TM-align (https://zhanggroup.org/TM-align/; accessed on 1 November 2021; [46]), and the TM-score normalized for the size of the latter.

### 2.3. Protein–Protein Docking

All protein 3D models used in this study were obtained using the I-TASSER web server (http://zhanglab/I-TASSER/; accessed on 1 November 2021 [45]). In order to select the inferred structure that most closely resembles the human ACE2 and the SARS-CoV-2 structures, the alternative 3D models were aligned, using TM-align (https://zhanggroup.org/TM-align/; accessed on 1 November 2021 [46]) with the human ACE2 structure (1R42-ttps://www.rcsb.org/structure/1R42; accessed on 1 November 2021) and SARS-CoV-2 S protein structure (6VYB, chain B-https://www.rcsb.org/structure/6VYB; accessed on 1 November 2021), respectively, and the one with the highest TM-score selected. Structure 1R42 was chosen because it is described as being the native Human protein and was not reported as a complex with inhibitors or other proteins. It should be noted that, although structure 1R42 is from 2004, it is almost identical to the most recently published ACE2 structure (7U0N; the crystal structure of chimeric omicron RBD complexed with human ACE2). Indeed, the TM-align score for the two structures is 0.99, where 1.00 indicates a perfect match. Structure 6VYB, from 2020, was used because it is the one reported in the article that established ACE2 as the SARS-CoV-2 S protein receptor [14]. Protein–protein docking inferences were made using the HADDOCK2.2 (https://alcazar./HADDOCK2.2/; accessed on 1 November 2021; [47]) web server. For both the S and ACE2 proteins, the active (the three neighboring sites of each active residue, or their orthologs identified based on MUSCLE [48] protein alignments) reported by Lan et al. [18] and passive sites (automatically defined by HADDOCK) were used. The cluster with the lowest Z-score was chosen, and for each cluster model interface residues identified using PDBePISA (https://www.ebi.ac.uk/pdbe/pisa/; accessed on 1 November 2021). The docking structure with the highest number of interfacing residues at the S protein was chosen, and in the case of a tie, the structure with the highest number of interfacing residues at the ACE2 receptor was chosen. Residues that appeared as interfacing in more than 90% of the inferences here regarding both the effect of SARS-CoV-2 mutations (12 datasets) and the binding of SARS-CoV-2 to the ACE2 receptor (13 datasets) from species known to be infected by this CoV, were labelled as core residues.

### 2.4. S protein: Sites under Positive Selection

All SARS-CoV-2 complete S gene sequences available at the ViPR’s *Coronaviridae* database were obtained. Given the very large number of sequences available, 100 random subsets were created that must have 20 sequences from the year 2019, 60 sequences from 2020 and 60 from 2021. The rationale for such a design is that the sequencing effort is not equal in the three years and, thus, randomly choosing sequences without constraints could lead to uninformative datasets made mostly by sequences from a given year or month. Due to computational constraints, from each of the 100 datasets, a set of 60 S sequences was chosen, and positively-selected amino acid sites identified using both FUBAR and codeML, as implemented in the Integrated Positively Selected Sites Analyses (IPSSA; [49]) Compi Pipeline. A Docker image is available for IPSSA at the pegi3S Bioinformatics Docker Images Project (https://pegi3s.github.io/dockerfiles/; accessed on 1 November 2021; [37]). In order to avoid false positives, only sites that were identified as positively selected in at least 25% of the replicas are considered.

### 2.5. Phylogenetic Inferences

A Bayesian phylogenetic tree was inferred for the CoVs S gene, using MUSCLE as the alignment algorithm [50] and MrBayes [51], as implemented in ADOPS (Automatic Detection Of Positively Selected Sites; [52]). A Docker image is available for ADOPS at the pegi3S Bioinformatics Docker Images Project (https://pegi3s.github.io/dockerfiles/; accessed on 1 November 2021; [37]). When using ADOPS, nucleotide sequences are first translated, and the translated sequences aligned, as implemented in T-Coffee [53]. The amino-acid alignment is then used as a guide to obtain the corresponding nucleotide alignment. For the phylogenetic reconstruction, only codons with a support value above two were used. The general time-reversible model (GTR) of sequence evolution was selected, allowing for among-site rate variation and a proportion of invariable sites. Third codon positions were allowed to have a gamma distribution shape parameter different from that of first and second codon positions. Two independent runs of 1,000,000 generations with four chains each (one cold and three heated chains) were performed. The potential scale reduction factor for every parameter was about 1.00 showing that convergence had been achieved. Trees were sampled every 100th generation, and a burn-in of 25% was used. The remaining trees were used to compute the Bayesian posterior probability values of each clade of the consensus tree.

### 2.6. Large Scale Predictive Model

From the sequences obtained in Section 2.1, we selected those ACE2 sequences from species that have been shown to be infected by SARS-CoV-2 (*Homo sapiens*-NP_001358344.1; *Felis catus*-NP_001034545.1; *Mustela putorius furo*-XP_004758942.1; *Macaca fascicularis*-XP_005593094.1; *Macaca mulatta*-XP_014982444.2; *Rousettus aegyptiacus*-XP_015974412.1; *Mesocricetus auratus*-XP_005074266.1; *Peromyscus maniculatus bairdii*-XP_006973269.1; *Callithrix jacchus*-XP_008987241.1; *Sus scrofa*-XP_020935033.1; *Mus musculus*-ENSMUSP00000073626; *Neovison vison*-ENSNVIP00000026442; and *Panthera leo*-ENSPTIP00000018286). The sequences were then aligned using MUSCLE [50]. The chemical properties of the ACE2-interfacing residues from species shown to be infected by SARS-CoV-2 were assessed, according to the classification of [54]. Only ACE2 amino acids that never appeared as non-interfacing in our inferences were considered.

## 3. Results

### 3.1. B Lineage CoVs That Bind to ACE2 Show a Distinctive Amino Acid Pattern

Visual inspection of the S protein alignment from species that are able and not able to bind ACE2, revealed a pattern that can be used to predict whether a given CoV is able or not to bind the ACE2 receptor. The pattern is CYX(6)GX(3)T[^V] and corresponds to positions 488–501 of SARS-CoV-2 S protein. This pattern is present in all CoVs known to use ACE2, with the exception of HCoV-NL63, and is absent in all CoVs not able to bind the ACE2 receptor. The 81 S protein sequences deposited in VIPR and NCBI that show this pattern were used to perform a phylogenetic analysis, where sequences representative of other β-coronaviruses lineages were also included (Appendix A). As can be seen, the CYX(6)GX(3)T[^V] pattern is a marker for the B lineage.

An approach based on the inferred S protein structure of CoVs can also be used to infer which species likely bind the ACE2 receptor. Indeed, species known to bind the ACE2 receptor show a higher TM-align score (see Material and Methods) than species not able to bind ACE2, when using the SARS-CoV-2 S protein RBM region as the reference (438–506). The average TM-score is 0.904 for species that are known to bind to ACE2, and 0.614 for those known not to bind the ACE2 receptor (Figure 1a). The two sets of values are statistically different (Figure 1b; N = 23; Non-parametric Mann–Whitney U test; *p* < 0.001).

### 3.2. More Recent SARS-CoV-2 Variants Are Inferred to Have Replaced the Old Ones Because They Are Able to Bind with Higher Affinity to the ACE2 Receptor

The global predicted number of interfacing residues (the sum of the number of S protein- and ACE2-interfacing residues) is always higher for SARS-CoV-2 S variant strains than for the original strain from 2019, although the number of ACE2- (Figure 2a) or S- (Figure 2b) interfacing residues can be lower than that inferred for the original strain from 2019. On average, the global predicted number of interfacing residues is 75.3 and 81.4 for variants of interest and concern, respectively, but there are no statistically significant differences between the two datasets (non-parametric Mann–Whitney U test; *p* > 0.05).

More recent SARS-CoV-2 variants seem to have a higher global predicted number of interfacing residues than older ones, suggesting that new variants replace the old ones because they are able to bind with higher affinity the ACE2 receptor (Figure 2). This is likely the case, since the positive selection analyses performed here identified all S protein amino acid changes of VOIs and VOCs, with the exception of site 732, as being positively-selected amino acid sites (Table 2). Although FUBAR led to the identification of a larger number of positively-selected amino acid sites than codeML when considering only those sites that were identified in at least 25% of the 100 datasets analyzed, there are eight sites in common on the top 12 FUBAR and codeML list (Table 2). The amino acid variants present in variants of concern and interest likely represent the ones conferring the highest advantage, although it is conceivable that many other amino acid variants also offer some advantage. Indeed, in total, FUBAR and codeML identified 182 and 111 positively-selected sites, of which 11 and 7 in the RBM, respectively.

While preparing this work for publication, the Omicron variant replaced the Delta variant and became dominant in many countries. The Omicron S protein/ACE2 docking shows a higher number of interfacing residues than that obtained for the Delta S protein/ACE2, thus, once again suggesting that new variants replace the old ones because they are able to bind with higher affinity to the ACE2 receptor (Figure 2).

### 3.3. The Number of ACE2-Interfacing Residues Is Positively Correlated with Transmissibility Rates

As expected, given the results of the previous section, a statistically significant positive correlation was found between the total number of interfacing residues and transmissibility rates when using a non-parametric approach (Spearman’s ρ = 0.68873; *p* < 0.01), but not when using a linear approach (Pearson’s r = 0.534; *p* > 0.05; Figure 3). Moreover, a statistically significant positive correlation was found between the number of ACE2-interfacing residues and transmissibility rates when using both approaches (Spearman’s ρ = 0.729; *p* < 0.01; Pearson’s r = 0.699; *p* < 0.01; Figure 3). Nevertheless, no significant correlation was found between the number of S protein-interfacing residues and transmissibility rates (Spearman’s ρ = 0.059; *p* > 0.05; Pearson’s r = 0.139; *p* > 0.05). This observation suggests that the same S amino acid site may interact with more than one amino acid at the ACE2 receptor. Only this way can the number of interfacing residues increase for the ACE2 receptor without increasing for the S protein. When evaluating the impact of a given mutation on the original S sequence regarding the number of interfacing residues, it is observed that mutations in the RBM region and outside this region can cause a similar impact (average value: inside RBM = 44.6; outside RBM = 41.5, non-parametric Mann–Whitney U test; N = 10; *p* > 0.05).

### 3.4. The Effect of Individual Amino Acid Substitutions

The effect that a given S protein amino acid substitution had on the number of ACE2 interfacing residues is not cumulative, since it varied depending on the presence of other S protein amino acid substitutions (Figure 4). For instance, the predicted effect of the E484K and A701V mutations when present on the ancestral background is usually higher than their effect on the background of the variants where they are present, with the exception of the B.1.351 background. Indeed, the number of ACE2-interfacing residues for the variants in which they are present can be the same as that for the original SARS-CoV-2 strain from 2019 (B.1.526), but also the highest inferred value (B.1.351 or 20H/501Y.V2). This implies that it is not possible to predict the impact of a given mutation without knowing the background in which it occurs.

### 3.5. Large Scale Predictive Model 

There are 11 ACE2 residues that are here inferred to be S protein/ACE2-interfacing residues in more than 90% of the inferences made for SARS-CoV-2 variants/human ACE2, or reference SARS-CoV-2/ACE2 proteins from species confirmed to be infected by SARS-CoV-2 (Table 3). All ACE2 residues here identified as interfacing, with the exception of site 386, have also been identified by Lan et al. [18] who determined the crystal structure of the receptor-binding domain (RBD) of the S protein of SARS-CoV-2 bound to the human receptor ACE2. Nevertheless, we did not discard site 386 from our analyses since it could be an interfacing residue when SARS-CoV-2 variants are bound to human ACE2 or when SARS-CoV-2 binds the ACE2 receptor of non-human species. This is a possibility since site 386 was identified as interfacing in more than 90% of the inferences regarding SARS-CoV-2 variants/human ACE2 binding, and in more than 90% of the inferences regarding the binding of SARS-CoV-2 and non-human ACE2 receptors. It should be noted that there is always an alanine at site 386 in species that are infected by SARS-CoV-2, while this site is variable in species that are not. There are only four out of the 125 mammal species here analyzed that do not show an alanine at this site (*Suricata suricatta* (XP 029786256.1), *Sorex araneus* (XP 004612266.1), *Fukomys damarensis* (XP 010643477.1), and *Miniopterus natalensis* (XP 016058453.1)) but birds (16 out 56) and reptilians (4 out of 20) can also have an alanine at site 386. Therefore, site 386 is not being used to exclude all non-mammal animals. Moreover, there are four S protein residues here inferred to be interacting residues that have not been identified by Lan et al. [18] (403, 484, 494, and 495), and five residues identified by these authors that are not here identified as contacting residues (446, 453, 475, 500, and 502) in more than 90% of the inferences made for SARS-CoV-2 variants/human ACE2, or reference SARS-CoV-2/ACE2 proteins from species confirmed to be infected by SARS-CoV-2 (Table 3). Using molecular dynamics simulations, amino acid site 484 has been recently identified as well as being important for ACE2 binding [55].

Eight out of the 11 ACE2 residues here inferred to be S protein/ACE2-interfacing residues can be grouped under a shared chemical feature, according to [54] (see Table 4). The derived pattern is [YSTHKREDQNWC]XXX[RE]X[REQ][ED]XX[YWH][REQ] at sites 31–42, and [HK] at site 353. Nevertheless, site 353 only excludes non-mammals and the mammal *Fukomys damarensis*, and, thus, this pattern can be shortened to [YSTHKREDQNWC]XXX[RE]X[REQ][ED]XX[YWH][REQ] at sites 31–42, when starting with mammals only.

When looking at the presence of the pattern [YSTHKREDQNWC]XXX[RE]X[REQ][ED]XX[YWH][REQ] at sites 31–42, and [HK] at site 353, in 295 ACE2 sequences from 287 different species, 109 (37.1%) show a match (see Appendix A). Of these, 102 are mammals, six are birds, all Passeriformes, and one is a reptile. 102 out of 129 (79.1%) mammalian species here analyzed are inferred to be susceptible to SARS-CoV-2. Primates are the most prevalent group with 25 species, followed by carnivores with 21 species from felines to canines and mustelids. Other prevalent orders include cetaceans and artiodactyls, both with ten species. Additionally, five out of 13 bat species (38%) are also inferred to be infected by SARS-CoV-2. All species groups identified as possible SARS-CoV-2 reservoirs have been here identified as being infected by SARS-CoV-2. Phylogenetically, it is clear that the mammalian candidates for SARS-CoV-2 susceptibility are well dispersed through Marsupialia and various clusters of Placentalia, only being absent in the order Monotremata (Appendix A).

Our approach compares well with that used by Damas et al. [35], who identified 109 mammal species with at least a moderate chance of susceptibility to SARS-CoV-2. Of the 240 species that have been analyzed in both studies, 138 were identified in both studies as non-susceptible and only one (*Fukomys damarensis*) was identified as non-susceptible by us and as susceptible by Damas et al. [35] (Figure 5). Moreover, 52 species have been identified as susceptible in both studies. Nevertheless, 49 species identified by us as being susceptible to SARS-CoV-2 are not identified by Damas et al. [35] as being susceptible. Among the latter category are five out of 13 bat species, and 5 out of 5 mustelids that are putative SARS-CoV-2 reservoirs.

Significant statistical differences are observed when comparing the number of predicted ACE2-interface residues in confirmed susceptible and non-susceptible species (42.6 and 36.6, respectively, Non-parametric Mann–Whitney U test; *p* < 0.05; Figure 6 and Appendix A). Therefore, for those species that unexpectedly show the above mentioned ACE2 pattern, protein docking inferences can be performed to confirm the status of the species as being susceptible to SARS-CoV-2. Nevertheless, the docking approach is much more time-consuming, and as such it is unfeasible to analyze a large number of species in this way. As can be seen in Appendix A, of all species confirmed to be susceptible to SARS-CoV-2, *Homo sapiens* presents the lowest number of ACE2-interfacing residues, namely 33. This observation suggests that the original SARS-CoV-2 strain was ill-adapted to humans, as expected for a species that recently crossed species boundaries. Indeed, as shown in Figure 2, the predicted number of interfacing residues has been increasing as new SARS-CoV-2 variants arise and get to high frequency due to positive selection (see previous section).

## 4. Discussion

Knowing which kind of receptor is used by a given CoV is important for predicting transmissibility rates using in silico approaches, as here performed. For five out of the seven CoVs known to infect humans, the protein receptor that is used has been identified. In three cases is ACE2, in one case dipeptidyl peptidase 4 (DPP4), and in another amino peptidase N (APN) [2]. Here, we have focused our attention on the ACE2 receptor because this seems to be the one mostly used in humans, and also because it is the one used by SARS-CoV-2.

The amino acid pattern CYX(6)GX(3)T[^V] (positions 488–501 of SARS-CoV-2 S protein) here identified can be used to predict which β-CoVs use the ACE2 receptor. Species known not to use the ACE2 receptor never present this pattern. In order to test whether this pattern is a marker for a given lineage, all S protein sequences available at NCBI and VIPR showing the pattern were retrieved. Phylogenetic analyses show that all sequences showing this pattern belong to the B lineage, and, thus, most likely, the amino acid pattern CYX(6)GX(3)T[^V] is a marker for this lineage.

The pattern here described performs better than the one presented by Carvalho and Alves [15], namely the hydrophobic sequence YGFY. It should be noted that this sequence is not present in SARS-CoV-2, and the Bat SARS-like CoV Rs7327, both known to use the ACE2 receptor. The YGFY sequence cannot be changed to accommodate these cases, since the Bat SARS-like CoV Rs7327 and the Bat CoV BM48–31 show the same sequence (YGFF) but one uses the ACE2 receptor and the other does not [16].

There is, however, one case where a CoV uses the ACE2 receptor, but that does not present the pattern here derived, namely HCoV-NL63, an endemic human CoV. This species employs a completely different method of binding to ACE2, such as more than one RBD region in its S protein [56]. CoVs that use the ACE2 receptor, with the exception of HCoV-NL63 which does not belong to the B lineage, show higher TM-scores of their RBM when aligned with the same region of SARS-CoV-2 S protein than those that do not use the ACE2 receptor.

This study found a correlation between the number of ACE2-interfacing residues and the transmissibility of the variants, suggesting that the increase in transmissibility may be linked to a better fit or more stable interaction of the S protein with the ACE2 receptor. This is surprising since transmissibility rates are difficult to estimate accurately because of the preventive measures imposed by governments to limit the spread of SARS-CoV-2. Every time a new VOC/VOI emerges, the number of interfacing residues increases. VOC/VOI likely increase in frequency due to positive selection, since all sites showing amino acid changes at the S protein of VOC/VOI, with the exception of site 732, have been identified as positively-selected amino acid sites. These results are also surprising because carbohydrate binding, not taken into account here, is also known to be important for SARS-CoV-2 binding as well (see for instance [19]). Moreover, it is known that two trimeric S protein structures bind an ACE2 homodimer, and here we look at the binding of a single S protein to a single ACE2 protein. While the use of models with higher complexity than the one presented here may reveal a better fit to the estimated transmissibility rates, the results here obtained show that a better fit or more stable interaction of the S protein with the ACE2 receptor is a main factor determining transmissibility rates.

Like in Damas et al. [35], cetaceans, primates, bovines, and felines are generally regarded as candidates for susceptibility. In contrast, the method here described considered also as candidates, mustelids, equids, rodents, and in particular, murids and otariids. The average score attributed by Damas et al. [35] to these groups was very low. *Mustela putorius furo* has been a confirmed SARS-CoV-2 susceptible species and the Denmark marten massacre show mustelids as a concern group [57], corroborating the results here shown. It is of note that the mammalian species inferred to be susceptible to SARS-CoV2 infection do not form a monophyletic group. Additionally, seven non-mammals were also scored positively here: six birds and one reptile. Notably, by applying the protein-docking protocol, the reptile *Zootoca vivipara*, who was not present in the Damas et al. [35] dataset, as well as *Ficedula albicolis*, *Serinus canaria,* and *Camarhynchus parvulus* (birds), show all core residues of both ACE2 and SARS-CoV-2’s S protein as interfacing. Our method also lends a stronger support to the origin of SARS-CoV-2 in bats, with five species being deemed as candidates for susceptibility. The simplistic prediction model here used can be applied to a large number of species (its greatest advantage), but it is likely that many other factors influence SARS-CoV-2 cross-species transmission. In order to address the performance of the simple model here presented, in comparison with more complex models, a much larger number of observations on species that can be or not be infected by SARS-CoV-2 is, however, needed.

As here demonstrated, useful predictions regarding the within- and between-species SARS-CoV-2 transmissibility can be made using relatively simple in silico approaches. It is of interest, but outside the scope of this article, to understand whether a similar approach can be used for other viruses.

## Figures and Tables

**Figure 1 viruses-14-01565-f001:**
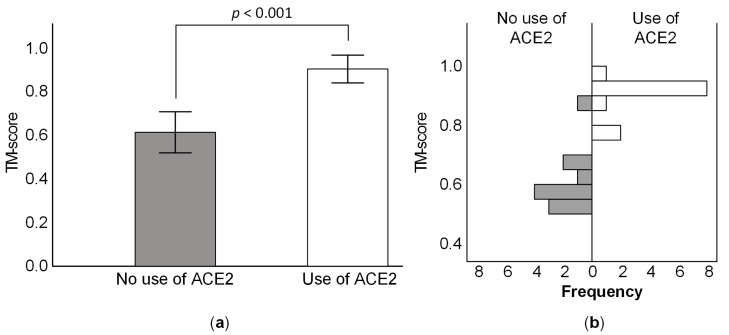
**The** TM-score for the RBM of species of coronavirus that are able or unable to use the ACE2 protein for host cell entry, when using as reference SARS-CoV-2. The score is normalized for the length of SARS-CoV-2’s RBM. (**a**) Average TM-score for the two groups; error bars represent one standard deviation. (**b**) Distribution of TM-scores for the two groups.

**Figure 2 viruses-14-01565-f002:**
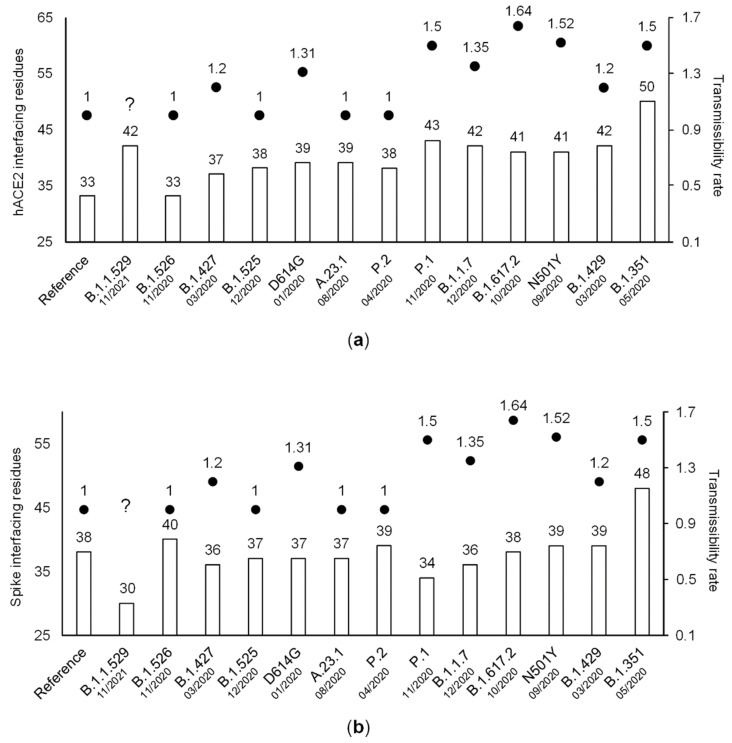
The number of hACE2 (**a**) and Spike (**b**) interfacing residues of SARS-CoV-2 variants and date of appearance, with their respective documented or expected transmissibility rate (dots). Results ordered by total number of interfacing residues.

**Figure 3 viruses-14-01565-f003:**
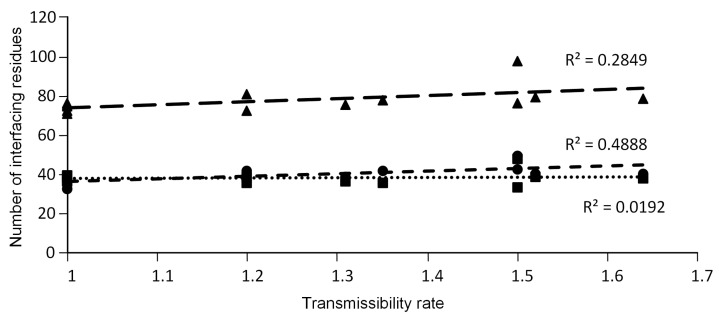
Correlation between the total (triangles), ACE2 (circles) and S (squares) number of interfacing residues and the expected/documented transmissibility rate of SARS-CoV-2 variants.

**Figure 4 viruses-14-01565-f004:**
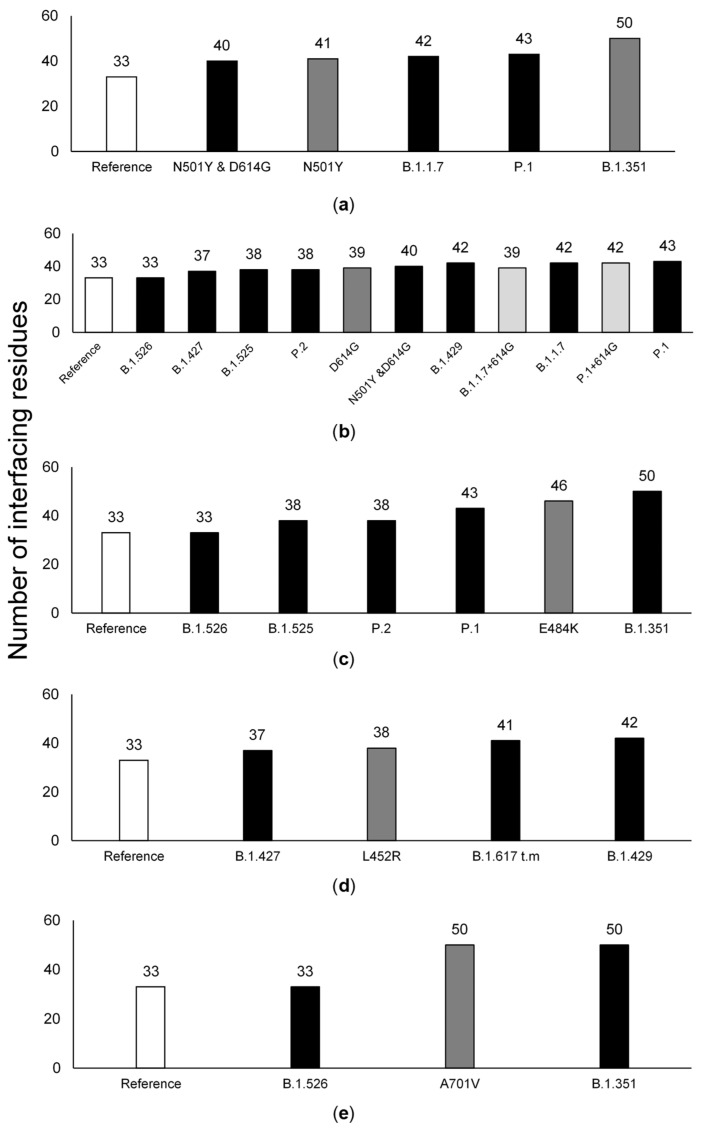
Effect of the point mutations N501Y (**a**), D614G (**b**), E484K (**c**), L452R (**d**), and A701V (**e**) to the ACE2 number of interfacing residues between hACE2 and S protein when applied to the reference sequence (grey), in tandem with variant-defining groups of mutations (black) or added to existing variants (light grey).

**Figure 5 viruses-14-01565-f005:**
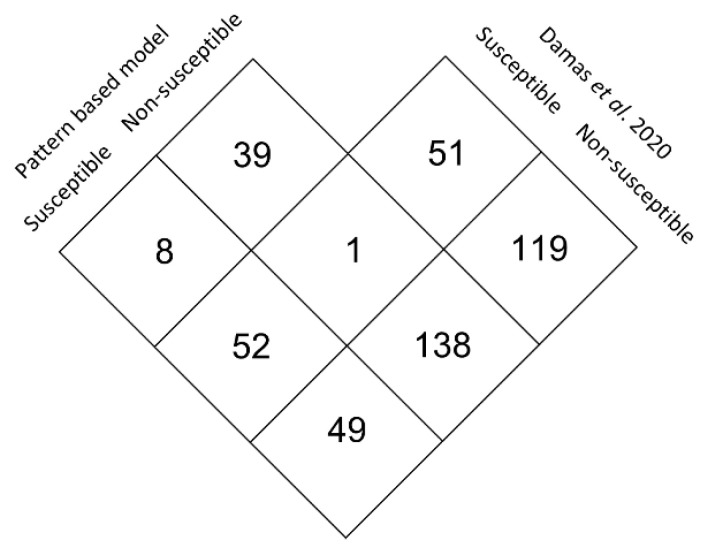
Comparison of the results obtained when using the [YSTHKREDQNWC]XXX[RE]X[REQ][ED]XX[YWH][REQ] at sites 31–42, and [HK] at site 353 pattern, and the results of Damas et al. [35].

**Figure 6 viruses-14-01565-f006:**
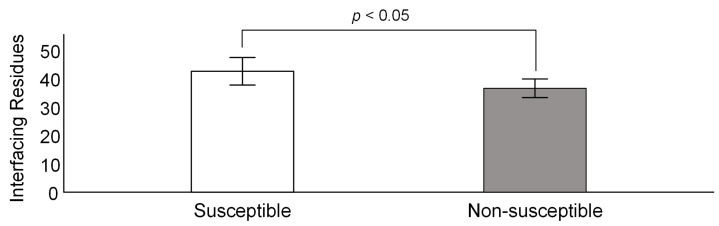
Mean number of ACE2-interfacing residues for susceptible (white) and non-susceptible (grey) SARS-CoV-2 species. Error bars represent one standard deviation.

**Table 1 viruses-14-01565-t001:** List of VOCs and VOIs identified by their Pango lineage identifier and their respective S protein mutations.

Variants	WHO Label	Mutations
A.23.1 *	-	F157L; V367F; Q613H; P681R
B.1.1.7 *	Alpha	del69–70HV; del144Y; N501Y; A570D; P681H; T716I; S982A; D1118H
B.1.351 *	Beta	D80A; D215G; K417N; E484K; N501Y; A701V
B.1.427 **	Epsilon	L452R; D614G
B.1.429 **	Epsilon	S13I; W152C; L452R; D614G
B.1.525 **	Eta	A67V; del69–70 HV; del144 Y; E484K; D614G; Q677H; F888L
B.1.526 **	Iota	L5F; T95I; D253G; S477N; E484K; D614G; A701V
P.1 *	Gamma	L18F; T20N; P26S; D138Y; R190S; K417T; E484K; N501Y; H655Y; T1027I
P.2 **	Zeta	E484K; D614G; V1176F
B.1.617.2 ***	Delta	T19R; T95I; G142D; E156-; F157-; R158G; L452R; T478K; D614G; P681R; D950N
B.1.1.529 ***	Omicron	A67V; H69del; V70del; T95I; G142D; V143del; Y144del; Y145del; N211del; L212I; ins214EPE; G339D; S371L; S373P; S375F; K417N; N440K; G446S; S477N; T478K; E484A; Q493R; G496S; Q498R; N501Y; Y505H; T547K; D614G; H655Y; N679K; P681H; N764K; D796Y; N856K; Q954H; N969K; L981F

* Pangolin [42]; ** CDC [43]; *** GISAID [44].

**Table 2 viruses-14-01565-t002:** SARS-CoV-2 S protein positively-selected sites identified in more than 25% of the codeML or FUBAR runs.

Method	25% < PSF ^1^ < 75%	PSF ^1^ >75%
codeML FUBAR	5, 681, 677 95, 732, 494, 138, 18, 26, 477, 681	- 5, 484, 501, 677

^1^ Frequency of runs showing positively-selected amino acids sites.

**Table 3 viruses-14-01565-t003:** S and ACE2 core residues (interfacing residues in more than 90% of the analyses here made) inferred from the modelling of species known to be infected by SARS-CoV-2, and SARS-CoV-2 variants. For ACE2, the residue number is based on the reference hACE2 protein sequence.

Protein	Residue Number
ACE2	30, 31, 34, 35, 37, 38, 41, 42, 353, 354, 386
S	403, 417, 449, 455, 456, 484, 486, 487, 489, 493, 494, 495, 496, 498, 501, 505

**Table 4 viruses-14-01565-t004:** Amino acid variation at interfacing ACE2 core residues and identified common chemical characteristics. Lack of a common characteristic signifies a very diverse site, with the exception of site 386, which is a conserved one.

Residue Number	Variation	Common Chemical Characteristic
30	[ADEN]	-
31	[EKNT]	Polar
34	[HLQSTVY]	-
35	[ER]	Big hydrophilic charged polar
37	[EQ]	Big hydrophilic polar
38	[DE]	Negatively charged polar
41	[HY]	Aromatic polar
42	[EQ]	Big hydrophilic polar
353	[HK]	Hydrophobic positively charged polar
354	[DGHNQR]	-
386	A	-

## Data Availability

Data is contained within the article or Appendix A.

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
