# Peer review of "Predictive Models of within- and between-Species SARS-CoV-2 Transmissibility"

_viruses, 2022, doi:10.3390/v14071565_

Round 1

Reviewer 1 Report

The work by Soares et al studies the roles of SARS-CoV-2 S and ACE2 interfacing residues in defining the SARS-CoV2 between and within species transmissibility. They performed in silico studies to prove the correlation between interfacing residues and transmissibility rates. The study provides a piece of additional information on the transmissibility of the SARS CoV-2 virus than published in previous reports. 

The work is carried out carefully and well presented. The introduction and discussion part is explained in detail.

I have a few minor points to consider

1. In the figure2 transmissibility rate is presented in commas instead of decimal

2. Please mentioned the limitations of the predicted model and study in the discussion part. 
